# Analysis of predicted factors for bronchoalveolar lavage recovery failure: An observational study

**Masafumi Shimoda**👁*, **Yoshiaki Tanaka, Kozo Morimoto, Taro Abe, Reina Asaga, Kei Nakajima, Ken Okamura, Kozo Yoshimori, Ken Ohta**👁

Respiratory Disease Center, Fukujuji Hospital, Japan Anti-Tuberculosis Association (JATA), Kiyose City, Tokyo, Japan

* shimodam@fukujuji.org

**Data Availability Statement:** All relevant data are within the paper and its Supporting information files.

## Abstract

### Background

The bronchoalveolar lavage (BAL) recovery rate should generally be more than 30% for effective diagnosis. However, there have been no reports investigating a target bronchus for BAL, and the cause of BAL recovery failure is uncertain. Therefore, this study detected predictive factors for BAL recovery failure through investigations on a target bronchus for BAL by using a 3D image analysis system. Therefore, this study detected predictive factors for BAL recovery failure.

### Materials and methods

We retrospectively collected data from 338 adult patients who underwent BAL procedures at Fukujuji Hospital from June 2018-March 2022. Factors correlated with the BAL recovery rate were detected. Furthermore, the patients were divided into the failure group (recovery rate <30%; 36 patients) and the success group (recovery rate $\geq$30%; 302 patients), and data were compared between the two groups by analysing the target bronchus by using a 3D image analysis system.

### Results

The patients in the failure group were older (median 74.5 years old [IQR 68.0–79.0] vs. median 70.0 years old [IQR 59.0–76.0], $p$ = 0.016), more likely to be male (n = 27 [75.0%] vs. n = 172 [57.0%], $p$ = 0.048), more likely to have COPD (n = 7 [19.4%] vs. n = 14 [4.6%], $p$ = 0.003), and more likely to perform a target site of BAL other than the middle/lingual lobe (n = 11 [30.5%] vs. n = 35 [11.6%], $p$ = 0.004) than those in the success group. The area of the bronchial wall was positively related to the recovery rate (r = 0.141, $p$ = 0.009), and the area of the bronchial wall in the failure group was lower than that in the success group (median 10.5 mm$^2$ [interquartile range (IQR) 8.1–14.6] vs. median 14.5 mm$^2$ [11.4–19.0], $p$<0.001).

**Funding:** The authors received no specific funding for this work.

**Competing interests:** The authors have declared that no competing interests exist.

**Abbreviations:** BAL, bronchoalveolar lavage; CI, confidence interval; COPD, chronic obstructive pulmonary disease; CT, computed tomography; $D_{LCO}$, carbon monoxide diffusing capacity; $FEV_1$/VC, forced expiratory volume in 1 second; $FEV_1$, forced expiratory volume in 1 second; IQR, interquartile range; ROC, receiver operating characteristic.

## Conclusion

The study shows that a thin bronchial wall, COPD, and a target site of BAL other than the middle/lingual lobe were identified as the predicted factors for BAL recovery failure. The weakness of the bronchial wall might cause bronchial collapse during the BAL procedure.

## Background

Bronchoalveolar lavage (BAL) is an established diagnostic tool for interstitial lung and infectious bronchopulmonary diseases [1]. Generally, the low BAL recovery rate is a poor study, and the BAL recovery rate should be more than 30% for an effective diagnosis [1, 2] because a total volume of retrieved fluid less than 30% may provide a misleading cell differential [3]. The predicted factors for a low BAL recovery rate were reported, such as male sex, elderly age, smoking history, chronic obstructive pulmonary disease (COPD), performing BAL at bronchi other than the middle lobe or lingula, and low forced expiratory volume in 1 second ($FEV_1$) divided by forced vital capacity ($FEV_1$/FVC) [1, 2, 4, 5]. However, few reports have compared recovery rates of <30% and recovery rates of ≥30% [5]; there have been no reports investigating a target bronchus for BAL. The common cause of BAL recovery failure might be the collapse of the bronchus [4], and we hypothesised that weakness of the bronchial wall might be related to BAL recovery failure. A 3D image analysis system can evaluate a target bronchus for BAL to calculate the area of the bronchial wall [6, 7]; Therefore, this study demonstrated that predicted factors for a BAL recovery rate of less than 30% could be detected by using a 3D image analysis system to investigate a target bronchus for BAL.

## Materials and methods

### Study design and setting

We retrospectively collected the data of 372 adult patients (age ≥18 years old) who underwent BAL procedures at Fukujuji Hospital from June 2018 to March 2022. The flowchart of the study is shown in Fig 1. A total of 338 patients with an available thin-slice thoracic CT scan were reviewed, excluding 31 patients whose data could not be analysed using a 3D image analysis system because they did not undergo thin-slice computed tomography (CT) scans and 1 patient with an unknown target site for BAL. Two patients were excluded because their BAL procedures could not be finished because of sputum obstruction to an instrumentation channel of the bronchoscope or because of wedges coming off.

Furthermore, the patients were divided into two groups based on the BAL total recovery rate, and the data were compared between the two groups. Data regarding symptoms, laboratory test results, radiological findings, and other relevant findings were collected. The study was approved by the Institutional Review Board of Fukujuji Hospital, the requirement for patient consent was waived because the study did not include any identifiable information for patients, and we applied the opt-out method. The decisions made by this board were based on and in accordance with the Declaration of Helsinki (Study number: 22001).

### Definition

The recovery rate was calculated as the ratio of the amount of recovery fluid after instillation to the amount of extracted fluid. The BAL failure group was identified as having a total recovery

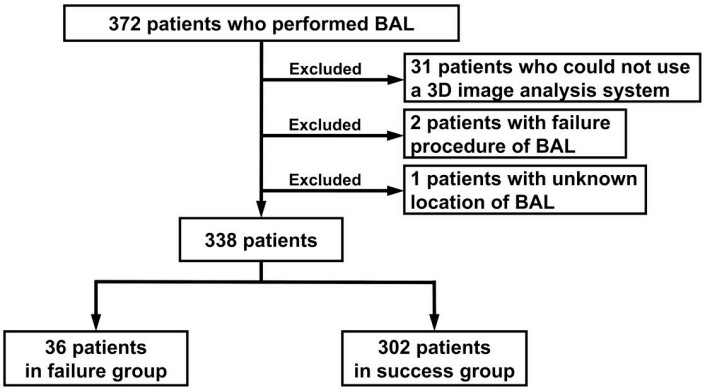

**Fig 1. Flowchart of the study.** BAL bronchoalveolar lavage fluid.

rate of less than 30%, which is considered a deterioration for an effective diagnosis in patients with interstitial lung disease [1, 2].

The area of the bronchial wall, the area of the bronchial lumen, and the lung volume affiliated bronchus, which is a target site for BAL, were calculated from CT images using a SYNAPSE VINCENT volume analyser (FUJIFILM Medical Co., Ltd., Tokyo, Japan), which is a 3D image analysis system; the system is very useful for evaluating respiratory function, surgical simulation function, virtual bronchoscopic navigation, and other parameters [8–10]. The outer and inner diameters of the bronchus were measured by lung analysis and were analysed at 5 points near the target bronchus orifice. The average of those 5 data points was used (Fig 2A). The area of the bronchial lumen was calculated as the area of the oval (area of the bronchial lumen = [major axis length of inner diameter]×[minor axis length of inner diameter]×3.14÷4). The area of the bronchial wall was calculated using the following formula: the area of the bronchial wall = (major axis length of outer diameter)×(minor axis length of outer diameter)×3.14÷4-(the area of the bronchial lumen). The lung volume was calculated by lung resection analysis (Fig 2B), which can analyse the lung volume dominated by a designated bronchus.

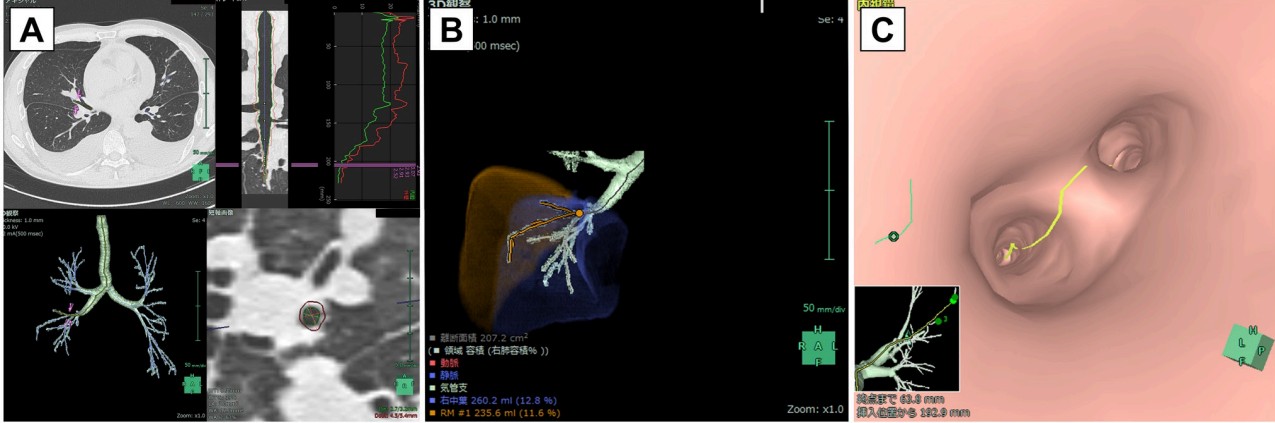

**Fig 2. A 3D image analysis system using a SYNAPSE VINCENT volume analyser calculated the area of the bronchial wall and bronchial lumen based on the lung analysis (A), lung volume affiliated bronchus in which BAL was performed based on the lung resection analysis (B), and bifurcation numbers of bronchus based on bronchoscopy simulator (C).**

A target site of the lung segment for performing BAL was divided into (1) the middle and lingual lobes, which are the usual lung segments for performing BAL, and (2) other segments, including the right upper and lower lobes, left segmentum apicoposterius and anterius, and left lower lobes. The bifurcation numbers of a distal bronchus from a target site for BAL were detected using the bronchoscopy simulator (Fig 2C). The bronchoscopy simulator can demonstrate the bronchial pathway to the peripheral lesion and observe the bifurcation of the bronchus on the target bronchus.

## BAL procedure

Bronchoscopy was performed under pharyngeal anaesthesia with 2% xylocaine solution and an intravenous premedication consisting of 1–5 mg midazolam as the sedative and/or 17.5–35 mg pethidine as the analgesic in a routine manner. The premedication doses were made as deemed appropriate by the handling physician. The bronchoscope was inserted transorally, and 2% xylocaine solution was sprinkled through the instrumentation channel of the bronchoscope. Oxygen humidified via the nasal tube was given during the examination, and the oxygen saturation was controlled with pulsoxymetry. To conduct BAL, the tip of the bronchoscope was placed into the wedge position in a lobe/segment/subsegmental bronchus. BAL was performed with three aliquots of 50 ml physiological saline at room temperature; using a method common in Japan, the saline was gradually instilled and then gently suctioned back through an instrumentation channel [2]. The recovery rate of BAL from the 50-, 100-, and 150-mL injections were labelled the recovery rates of the 1st, 2nd, and 3rd aliquots, respectively.

## Statistical methods

All data were analysed and processed using EZR, version 1.53 [11]. The Mann–Whitney U test, Pearson's chi-squared test, and binomial logistic regression analysis were used for group comparisons. The Kruskal–Wallis test was used to compare data among 3 groups or more, and Bonferroni's correction was used for comparative testing. The correlated factors of the BAL recovery rate were detected based on Spearman's correlation analysis. The odds ratios were calculated. A receiver operating characteristic (ROC) curve was constructed and used to determine the cut-off values. The level of statistical significance was set at $p = 0.05$ (2-tailed).

## Results

In the baseline characteristics of the study, the median age was 70.5 years (interquartile range (IQR): 60.0–77.0), and there were 200 males (59.2%). The median BAL recovery rate was 51.0% (IQR: 39.3–60.0), including 15.0% (IQR: 10.0–19.0) of the 1st aliquot, 24.0% (IQR: 18.0–30.0) of the 2nd aliquot, and 37.0% (IQR: 29.0–43.0) of the 3rd aliquot. There were 36 patients (10.7%) in the failure group and 302 patients (89.3%) in the success group.

The comparisons between the failure group and the success group are shown in Table 1. Male sex, COPD, a target site of BAL other than the middle/lingular segment, the area of the bronchial wall, and median were associated with a low rate of BAL.

The correlations with the recovery rate of BAL were calculated (S1 Table). Age (r = -0.131, $p = 0.016$), the number of cigarettes smoked (pack-year) (r = -0.212, $p < 0.001$), and white blood cell count (r = -0.132, $p = 0.015$) were negatively related to the recovery rate. The area of the bronchial wall was positively related to the recovery rate (r = 0.141, $p = 0.009$). In contrast, the area of the bronchial lumen (r = 0.023, $p = 0.672$) and lung volume affiliated with a target bronchus for BAL (r = 0.003, $p = 0.952$) did not show a significant relationship.

Table 2 shows comparisons of the BAL recovery rate for affecting factors. The recovery rate was lower in patients who were males, had comorbidities, had COPD, and had bronchial

**Table 1. Comparisons between the failure group and the success group.**

|  | The failure group (n = 36) | The success group (n = 302) | p value |
|---|---|---|---|
| Age, median (IQR), years | 74.5 (68.0–79.0) | 70.0 (59.0–76.0) | 0.016 |
| Sex (Male/Female) | 27/9 | 172/130 | 0.048 |
| Having Comorbidity | 31 (86.1) | 266 (88.1) | 0.943 |
| COPD, n (%) | 7 (19.4) | 14 (4.6) | 0.003 |
| Bronchial asthma, n (%) | 7 (19.4) | 27 (8.9) | 0.071 |
| The number of cigarettes smoked, n (pack-year) [a] | 15.5 (0–30.9) | 8.9 (0–33.0) | 0.383 |
| Laboratory findings; WBCs, median (IQR), cells/μL | 7,785 (6,695–8,525) | 7,485 (6,000–9,133) | 0.556 |
| A target site of BAL other than the middle/lingual lobe, n (%) | 11 (30.5) | 35 (11.6) | 0.004 |
| The area of the bronchial lumen, median (IQR), mm$^2$ | 8.0 (5.9–12.5) | 9.4 (7.1–12.6) | 0.418 |
| The area of the bronchial wall, median (IQR), mm$^2$ | 10.5 (8.1–14.6) | 14.5 (11.4–19.0) | <0.001 |

IQR interquartile range, COPD chronic obstructive pulmonary disease, BAL bronchoalveolar lavage fluid

[a]: n = 326

asthma. Past smokers showed a lower recovery rate than never smokers, while current smokers did not show a significant difference from never smokers or past smokers. Regarding a target site for performing BAL, there was no significant difference in the recovery rate in the left-right lung and lobe/segment/subsegment bronchus. The median recovery rate in the middle/lingual lobe was higher than that in the other lobes. The other lobes were the upper lobe/superior segment in 27 patients and the lower lobe in 19 patents.

Three factors for BAL recovery rate failure, including a target site of BAL other than the middle/lingual lobe, COPD, and an area of bronchial wall <10.6 mm$^2$, were analysed using binomial logistic regression analysis (Table 3). The cut-off value of the area of the bronchial wall for predicting BAL recovery rate failure was identified by using an ROC (Fig 3). The three factors were selected based on high odds ratios using Pearson's chi-squared test (S2 Table). Three factors were associated with a high risk of BAL recovery rate failure and with high odds ratios.

In addition, the area of the bronchial wall did not show significant relationships with COPD (having COPD 13.8 mm$^2$ [11.2–15.3] vs. no COPD 14.2 mm$^2$ [10.9–15.3], $p$ = 0.456), a target site of BAL (middle/lingual lobe 14.1 mm$^2$ [10.9–18.5] vs. other lobes 14.3 mm$^2$ [11.6–22.7], $p$ = 0.241), sex (male 13.8 mm$^2$ [10.4–17.6] vs. female 14.8 mm$^2$ [11.3–19.2], $p$ = 0.169), or age (>71 years old median 14.1 mm$^2$ [10.6–19.1] vs. ≤71 years old median 14.2 mm$^2$ [11.2–18.6], $p$ = 0.877) (S1 Fig).

## Discussion

This study shows that the thickness of the bronchial wall on a target bronchus for BAL by using a 3D image analysis system was detected as the predicted factor for BAL recovery rate failure, in addition to having COPD and a target site of BAL on other than the middle/lingual lobe. Previous studies report that COPD, a target site of BAL other than the middle/lingual lobe, male sex, and age are related to the BAL recovery rate [1, 2, 4, 5], similar to our data. In addition, only one study has reported on predicted factors for less than a 30% BAL recovery rate [5]. However, previous reports did not investigate a target bronchus for BAL, and no study has reported that a thin bronchial wall is related to the BAL recovery rate.

**Table 2. Factors affecting the recovery rate of BAL.**

| Variables | | Number of patients, n (%) | Recovery rate, % (IQR) | *p value* |
|---|---|---|---|---|
| Sex | Male | 200 (59.2) | 48.0 (37.5–58.0) | <0.001 |
| | Female | 138 (40.8) | 55.0 (46.3–61.0) | |
| Having comorbidity | Yes | 302 (89.3) | 51.0 (39.0–60.0) | 0.005 |
| | No | 36 (10.7) | 57.5 (50.8–63.3) | |
| COPD | Yes | 22 (6.5) | 38.0 (26.0–45.0) | <0.001 |
| | No | 317 (93.5) | 52.0 (41.0–61.0) | |
| Bronchial asthma | Yes | 35 (10.3) | 48.0 (34.3–55.0) | 0.034 |
| | No | 304 (89.7) | 51.0 (41.0–61.0) | |
| Smoking history | Never | 127 (37.5) | 55.0 (45.0–62.0) | 0.002* |
| | Past | 174 (51.3) | 48.5 (37.0–87.0) | |
| | Current | 37 (10.9) | 52.0 (41.0–57.0) | |
| Having symptom | Yes | 297 (87.9) | 51.0 (39.0–60.0) | 0.854 |
| | No | 41 (12.1) | 51.0 (44.0–60.0) | |
| A target site for performing BAL | | | | |
| Left-right | Right | 248 (73.2) | 51.0 (39.3–61.8) | 0.602 |
| | Left | 91 (26.8) | 51.0 (40.0–59.5) | |
| Segment | Middle/lingual lobe | 293 (86.4) | 51.0 (42.0–61.0) | 0.008 |
| | Other | 46 (13.6) | 47.0 (30.3–55.0) | |
| Bronchus | lobe | 14 (4.1) | 49.0 (41.0–56.0) | 0.148 |
| | Segment | 173 (51.0) | 55.0 (42.0–61.0) | |
| | Subsegment | 152 (44.8) | 49.0 (39.0–59.0) | |
| Bifurcation numbers of bronchus | Two | 274 (81.7) | 51.0 (39.0–60.3) | 0.815 |
| | Three or more | 62 (18.3) | 51.0 (41.3–59.0) | |
| Handling physician | Resident | 91 (26.8) | 49.0 (38.3–57.8) | 0.174 |
| | Senior doctor | 248 (73.2) | 52.0 (41.0–61.0) | |

IQR interquartile range, COPD chronic obstructive pulmonary disease

*: Patients with a past smoking history had a lower recovery rate of BAL than never smoker patients, which was significant after Bonferroni's correction.

The common cause of BAL recovery rate failure is the collapse of the bronchus [4]. In the BAL procedure, fluid is suctioned back with a negative pressure connected to the working channel of the bronchoscope. The risk of bronchial collapse can increase due to loss of elastic recoil and increasing compliance with the bronchus [2, 4], which might be related to weakness of the bronchial wall [2, 4]. Therefore, the fact that a thin bronchial wall was related to a low BAL recovery rate is very important, and it is hypothesised that weakness of the bronchial wall might be related to BAL recovery failure. Weakness of the bronchial wall is also seen in older individuals [12]. Both atrophy of the bronchial glands and mucosa and reduced compliance in

**Table 3. Binomial logistic regression analysis of the predictive factors for the BAL recovery rate failure.**

| | Odds ratio | 95% Confidence interval | | *p value* |
|---|---|---|---|---|
| | | Upper limit | Lower limit | |
| A target site of BAL other than the middle/lingual lobe | 4.11 | 1.72 | 9.80 | 0.001 |
| COPD | 6.21 | 2.09 | 18.5 | 0.001 |
| The area of the bronchial wall <10.6 mm$^2$ | 5.22 | 2.43 | 11.2 | <0.001 |

BAL bronchoalveolar lavage fluid, COPD chronic obstructive pulmonary disease

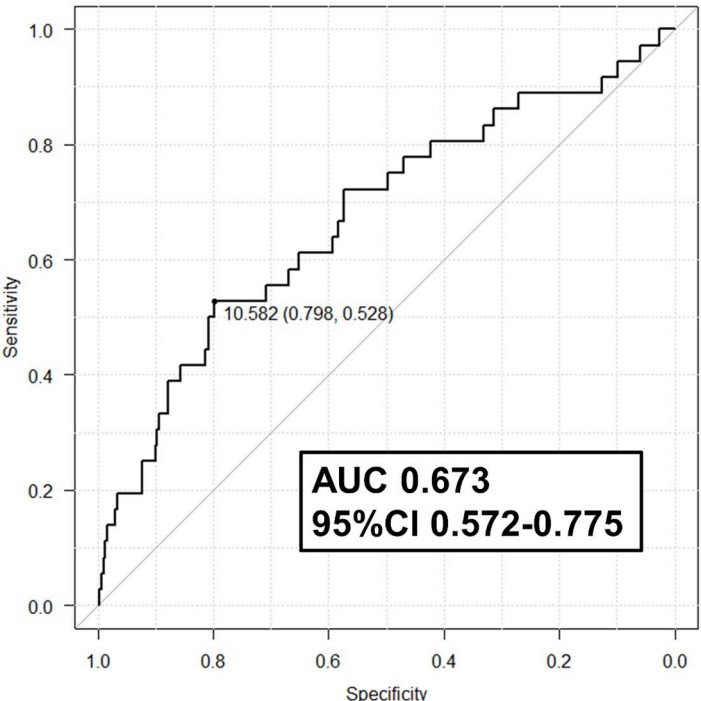

**Fig 3. ROC of the area of the bronchial wall for predicting BAL recovery rate failure.** The AUC was 0.673 (95% CI 0.572 to 0.775). The cut off value, which was decided by a point of maximum sensitivity and specificity, was 10.6 mm$^2$. ROC: receiver operating characteristic curve, AUC: area under the curve, CI: confidence interval.

the lung parenchyma are caused by ageing [2, 5, 13]. Therefore, elderly individuals may easily induce a collapse of the airway during BAL [5, 13].

According to previous reports, bronchial wall thickness is related to the severity of COPD [6, 7]. However, the bronchial wall in patients with COPD is thicker than that in normal subjects [14], and the high severity of COPD is associated with a thicker bronchial wall [6, 7]. This might seem to differ from our data, such that a lower area of the bronchial wall was related to BAL recovery failure. In our study, a thin bronchial wall, COPD, and a target site of BAL other than the middle/lingual lobe were independent predicted factors for BAL recovery failure based on binomial logistic regression analysis. There was no significant difference between the bronchial wall thickness and COPD. Therefore, there can be several causes for BAL recovery failure.

In patients with COPD, the recovered fluid is reduced to 10–40% of that of instilled patients [15], and emphysema might be related to a cause of low BAL recovery failure [4, 16]. However, an increasing lung volume due to emphysema might not be related to a low BAL recovery; indeed, our data showed no relationship between the BAL recovery rate and lung volume affiliated with the bronchus, which is a target site for BAL. A previous report suggests that a low BAL recovery rate may reflect larger airways rather than the alveolar compartment in patients with COPD [4]. Generally, airway obstruction in COPD is caused both by bronchiolitis and emphysema, and loss of lung elastic recoil and accompanying increased compliance are regarded as the pathophysiological characteristics of pulmonary emphysema [4].

Furthermore, our data and previous reports showed that the BAL recovery rate was lower in regions other than the middle/lingual lobe [2, 5]. The guidelines suggest that the target site

should be selected based on thin-slice CT rather than selecting the middle/lingual lobe [3], although that evidence has not been fully established [5]. The supine position of a patient during bronchoscopy might be related to the BAL recovery rate because the orifice of the middle lobe bronchus and lingula are located in areas that resist gravity [2]. Therefore, the BAL target site should be in the middle/lingual lobe on CT if abnormalities are present in the middle/lingual lobe [5]. Low recovery rates of BAL may not only lead to an inaccurate diagnosis but may also lead to an increase in adverse events [2, 17]; therefore, the BAL procedure should be avoided in bronchi with thin bronchial walls and in bronchi other than the middle/lingual lobe as much as possible.

This investigation has several limitations. The study was conducted retrospectively at a single centre. Analysis of the lung volume did not take into account adjusted data based on physiques such as body weight, body length, and body mass index. It could not be analysed in some patients by using the SYNAPSE VINCENT volume analyser. The attending physician chose a target site for performing BAL. BAL was performed mainly on the middle/lingual lobe if abnormalities were present on CT and was performed on the other lobe with abnormalities if the middle/lingual lobe did not have abnormal lesions. CT scans for prebronchoscopy are usually performed approximately 1–8 weeks before bronchoscopy; therefore, the analysis of CT scans might not reflect the condition of a patient on the day of inspection. Predicted factors such as age, male sex, and COPD might be confounding factors, while previous studies report that age, male sex, and smoking history are independent predictive factors [2, 5]. However, we could only analyse three factors or less using binomial logistic regression analysis because there were 36 patients with BAL recovery failure.

## Conclusion

This study shows the predicted factors for BAL recovery failure, such as a thin bronchial wall, COPD, and a target site of BAL other than the middle/lingual lobe. In particular, it is very important that a thin bronchial wall calculated in a SYNAPSE VINCENT volume analyser was related to the BAL recovery rate. It is considered that weakness of the bronchial wall might cause bronchial collapse during the BAL procedure.

## Supporting information

**S1 Fig.** The area of the bronchial wall did not show significant relationships with COPD (having COPD 13.8 mm2 [11.2–15.3] vs. no COPD 14.2 mm2 [10.9–15.3], p = 0.456) in S1A Fig, a target site of BAL (middle/lingual lobe 14.1 mm2 [10.9–18.5] vs. other lobes 14.3 mm2 [11.6–22.7], p = 0.241) in S1B Fig, sex (male 13.8 mm2 [10.4–17.6] vs. female 14.8 mm2 [11.3–19.2], p = 0.169) in S1C Fig, and age (>71 years old median 14.1 mm2 [10.6–19.1] vs. ≤71 years old median 14.2 mm2 [11.2–18.6], p = 0.877) in S1D Fig.
(TIF)

**S1 Table. Correlation with the recovery rate of bronchoalveolar lavage fluid.**
(DOCX)

**S2 Table. The odds ratio for the BAL recovery rate failure was analysed using Pearson's chi-squared test.**
(DOCX)

## Author Contributions

**Conceptualization:** Masafumi Shimoda.

**Data curation:** Masafumi Shimoda, Kozo Morimoto, Taro Abe, Kei Nakajima, Ken Okamura, Kozo Yoshimori.

**Formal analysis:** Masafumi Shimoda.

**Investigation:** Masafumi Shimoda.

**Methodology:** Masafumi Shimoda.

**Project administration:** Ken Ohta.

**Resources:** Masafumi Shimoda.

**Software:** Masafumi Shimoda.

**Supervision:** Yoshiaki Tanaka, Kozo Yoshimori, Ken Ohta.

**Visualization:** Masafumi Shimoda.

**Writing – original draft:** Masafumi Shimoda.

**Writing – review & editing:** Masafumi Shimoda, Yoshiaki Tanaka, Reina Asaga.

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
