## [Decision Letter · Decision Letter 0]

4 Aug 2022

PONE-D-22-18216Analysis of Predicted factors for Bronchoalveolar Lavage Recovery Failure: observational studyPLOS ONE

Dear Dr. Shimoda,

Thank you for submitting your manuscript to PLOS ONE. After careful consideration, we feel that it has merit but does not fully meet PLOS ONE’s publication criteria as it currently stands. Therefore, we invite you to submit a revised version of the manuscript that addresses the points raised during the review process.

We look forward to receiving your revised manuscript.

Kind regards,

Taeyun Kim

Academic Editor

PLOS ONE

Journal Requirements:

Additional Editor Comments (if provided):

* I suggest tone down of the Conclusion. The study only observed, found, or showed, rather than demonstrated.

* As the Reviewer pointed out, many parts of this paper overlap with the very recent study in the same country; Clin Respir J. 2022 Feb;16(2):142-151. Therefore, I suggest re-write their paper emphasizing the difference of your paper with that study;Maybe, using 3D image analysis and including target bronchus.

* The Introduction section seems poor in content. I can't easily agree why a new paper using 3D image analysis and separately analyzing target bronchus is needed. In addition, the readers may not easily understand why 30% is important and why investigating a target bronchus is important.

* Please fully explain what BAL recovery rate means; the % of extracted fluid after the instillation.

* Line 101-101; The correlated factors of the BAL recovery rate were detected based on Spearman’s correlation analysis; Please move Statistical analysis section.

* Line 164-165; I can't find the results reporting sensitivity, specificity, and accuracy.

* Line 228, I can't find any results related to ROC. Also, the cut-off value was different from 30%? which was validated in a previous study; Respiration 2007;74:553–557.

* Table 3; I think this is unnecessary. Given this study aims to compare the factors between failure and success group, I suggest showing the difference of baseline characteristics between those groups in Table 1.

* For readability, please separate Study design and setting into 2 or 3 paragraphs.

* For readability, please separate Definition section into 2 or 3 paragraphs.

* I wonder the difference between upper and lower segments. Are there any difference between upper, middle/lingular, vs. lower segments?

* Please do not just simply repeat the values that can be found in the Tables. For example, "the median age was 70.5 IQR 60-77" is unnecessary. Please briefly summarize the findings in Tables or Figures. For example, they could state; in univariable analysis, male sex, having comorbidity, COPD, asthma, other segment than middle/lingular, was associated with low rate of BALF.

* Line 248, I suggest not stating "first" and "demonstrate". This is an observational study. Several studies have reported the factors affecting BALF. Rather, emphasizing the differences of the authors' paper between previous things; 3D, target bronchus, or airway thickness.

* Line 254, please delete "to the best of our knowledge".

* Line 258-261. Please re-write this section as a separate paragraph, discussing why thin bronchial wall is related to a low BALF.

* Line 277-296. Too long. Please separate this section; one for COPD and another for elderly participants.

* Line 297-317. Too long and exhausting to follow.

* The supporting file, data.xlsx, seems unnecessary.

Reviewers' comments:

Reviewer's Responses to Questions

**Comments to the Author**

1. Is the manuscript technically sound, and do the data support the conclusions?

Reviewer #1: Yes

Reviewer #2: Yes

2. Has the statistical analysis been performed appropriately and rigorously? 

Reviewer #1: Yes

Reviewer #2: Yes

3. Have the authors made all data underlying the findings in their manuscript fully available?

Reviewer #1: Yes

Reviewer #2: Yes

4. Is the manuscript presented in an intelligible fashion and written in standard English?

Reviewer #1: Yes

Reviewer #2: No

5. Review Comments to the Author

Reviewer #1: My concern is that this overlaps substantially with reference 2 and I do not see what this paper adds to the literature as the findings were nearly identical.

The study design is good and statistics support the conclusion. If reference 2 had not been published, I would recommend for publication, but as it does not add to the literature I cannot recommend it.

Reviewer #2: Review report PONE-D-22-18216

« Analysis of Predicted factors for Bronchoalveolar Lavage Recovery Failure: observational study »

Dr Shimoda and al report the results of a retrospective study looking for predictors of poor bronchoalveolar lavage (BAL) recovery rate during bronchoscopy. A poor recovery rate was determined as below 30% of the instilled liquid.

Patients included were those, during the study period, who also had undergone thoracic CTscan allowing 3D image analysis of the bronchial tree.

Underlying conditions (COPD), tobacco use, age, lung lobe where the BAL was performed, cross-sectional area of the dependent bronchus and area of the bronchus wall were studied as potential predictors of BAL recovery rate.

The authors identified COPD, tobacco use history, BAL performed in another lobe than middle lobe or lingula and area of the bronchial wall as factors independently linked to the BAL recovery rate.

The authors used 3D image analysis to study the role of bronchial area and bronchial wall area; this makes the study very original.

I have few minor comments.

Though I am not a native English speaker, I wander if the manuscript doesn’t need English editing throughout.

Introduction and methods: The authors should indicate clearly that an inclusion criterion was the availability of a thin-slice thoracic CT scan.

I am not sure that waiving the need for patient consent, even in a retrospective study, is really in accordance with the Declaration of Helsinki.

6. PLOS authors have the option to publish the peer review history of their article (what does this mean?). If published, this will include your full peer review and any attached files.

Reviewer #1: No

Reviewer #2: No

---

## [Author Response · Author response to Decision Letter 0]

12 Aug 2022

Responses to the editor and the reviewers

Manuscript ID PONE-D-22-18216

“Analysis of Predicted factors for Bronchoalveolar Lavage Recovery Failure: A Case–Control Study”

All modifications have been highlighted in the revised manuscript.

To the editor and the reviewer:

Thank you very much for your constructive comments on our manuscript PONE-D-22-18216. We are pleased to hear that the reviewer found our study intriguing. In response to the reviewer’s requests and comments, we generated text and tables. We carefully studied all comments and made the necessary edits/modifications. The revised version of the manuscript has been edited by a professional English-editing service. The point-by-point responses are listed below. We hope that we have sufficiently addressed the issues raised by the reviewer.

Additional Editor Comments

* I suggest tone down of the Conclusion. The study only observed, found, or showed, rather than demonstrated.

Response: Thank you for your comment. We agreed your comment, and changed “demonstrats” to “shows” in abstract and on page 21, line 324.

* As the Reviewer pointed out, many parts of this paper overlap with the very recent study in the same country; Clin Respir J. 2022 Feb;16(2):142-151. Therefore, I suggest re-write their paper emphasizing the difference of your paper with that study;Maybe, using 3D image analysis and including target bronchus.

Response: Thank you for your comment. We believe that this study can provide new important information such as the thickness of the bronchial wall on a target bronchus for BAL by using a SYNAPSE VINCENT volume analyser was the predicted factors for BAL recovery rate failure. Therefore, we understand that this manuscript should emphasize it for the difference from other study. We edited in abstract, on page 5, lines 85-89, on page 16, line 249-page 17, line 252, and on page 20, lines 326-327.

* The Introduction section seems poor in content. I can't easily agree why a new paper using 3D image analysis and separately analyzing target bronchus is needed. In addition, the readers may not easily understand why 30% is important and why investigating a target bronchus is important.

Response: Thank you for your comment. We considered that the common cause of BAL recovery failure might be the collapse of the bronchus, and we hypothesised that weakness of the bronchial wall might be related to a BAL recovery failure. A 3D image analysis system can evaluate a target bronchus for BAL to calculate the area of the bronchial wall; therefore, we analyzed target bronchus by using a 3D image analysis system. We modified it on page 5, lines 85-89. 

Furthermore, a total volume of retrieved fluid less than 30% may provide a misleading cell differential. We added it on page 5, lines 78-79.

* Please fully explain what BAL recovery rate means; the % of extracted fluid after the instillation.

Response: Thank you for your comment. Recovery rate was calculated as ratio of recovery fluid amount after the instillation and extracted fluid amount. We added it on page 7, lines 121-122.

* Line 101-101; The correlated factors of the BAL recovery rate were detected based on Spearman’s correlation analysis; Please move Statistical analysis section.

Response: Thank you for your comment. We moved it to Statistical analysis section on page 10, lines 178-179.

* Line 164-165; I can't find the results reporting sensitivity, specificity, and accuracy.

Response: Thank you for your comment, and we apologize for a mistake. Our manuscript did not include sensitivity and specificity; therefore, we deleted it.

* Line 228, I can't find any results related to ROC. Also, the cut-off value was different from 30%? which was validated in a previous study; Respiration 2007;74:553–557.

Response: Thank you for your comment. I’m sorry for sentences that are difficult to understand. The ROC was analyzed for predicting BAL recovery rate failure (<30%) in the area of the bronchial wall. We provided the ROC in figure 3, and modified it on page 15, lines 222-224. 

* Table 3; I think this is unnecessary. Given this study aims to compare the factors between failure and success group, I suggest showing the difference of baseline characteristics between those groups in Table 1.

Response: Thank you for your comment. We agreed your ideas that the study aims were comparisons the factors between the BAL recovery failure and success groups. Therefore, Table 1 was delated, and the baseline characteristics between those groups were shown in Table 2 (Table 3 in initial version changed to table 2 in revise version).

* For readability, please separate Study design and setting into 2 or 3 paragraphs.

Response: Thank you for your comment. We modified to separate Study design and setting into 2 paragraphs.

* For readability, please separate Definition section into 2 or 3 paragraphs.

Response: Thank you for your comment. We modified to separate Definition section into 3 paragraphs.

* I wonder the difference between upper and lower segments. Are there any difference between upper, middle/lingular, vs. lower segments?

Response: Thank you for your comment. We compared a target site of the lung segment for performing BAL dividing into (1) the middle and lingual lobes, which are the usual lung segments for performing BAL, and (2) other segments, including the right upper and lower lobes, left segmentum apicoposterius　and anterius, and left lower lobes. We described it on page 8, line 142-page 9, line 145.

* Please do not just simply repeat the values that can be found in the Tables. For example, "the median age was 70.5 IQR 60-77" is unnecessary. Please briefly summarize the findings in Tables or Figures. For example, they could state; in univariable analysis, male sex, having comorbidity, COPD, asthma, other segment than middle/lingular, was associated with low rate of BALF.

Response: Thank you for your comment. We edited result section. Data which described in table were deleted, and we summarized the findings. Please confirm it in result section.

* Line 248, I suggest not stating "first" and "demonstrate". This is an observational study. Several studies have reported the factors affecting BALF. Rather, emphasizing the differences of the authors' paper between previous things; 3D, target bronchus, or airway thickness.

Response: Thank you for your comment. We agreed your ideas, therefore “is the first report to demonstrate” was changed to “shows” on page 16, line 249.

* Line 254, please delete "to the best of our knowledge".

Response: Thank you for your comment. We deleted it.

* Line 258-261. Please re-write this section as a separate paragraph, discussing why thin bronchial wall is related to a low BALF.

* Line 277-296. Too long. Please separate this section; one for COPD and another for elderly participants.

Response: Thank you for your comment. We re-write this section as a separate paragraph and discuss with elderly participants. Please confirm it on page 17, line 259-page 18, line 270. And, a description of a SYNAPSE VINCENT volume analyser such as “the system is very useful for evaluating a respiratory function, surgical simulation function, virtual bronchoscopic navigation, and other parameters” moved to methods on page 8, lines 129-131.

* Line 297-317. Too long and exhausting to follow.

Response: Thank you for your comment. We deleted sentence for male sex and handling physician, and it became shorter. Please confirm it.

* The supporting file, data.xlsx, seems unnecessary.

Response: Thank you for your comment. We agreed your ideas, therefore those files were deleted.

Reviewer #1: 

My concern is that this overlaps substantially with reference 2 and I do not see what this paper adds to the literature as the findings were nearly identical. The study design is good and statistics support the conclusion. If reference 2 had not been published, I would recommend for publication, but as it does not add to the literature I cannot recommend it.

Response: Thank you for your comments. Previous reports including reference 2 demonstrates that sex, age, bronchus used for the procedure, and having COPD are risk factors for BAL recovery failure, similar to our data. However, no report investigates a relationship between the area of the bronchial wall and BAL recovery rate, and it is importance that this study revealed the thickness of the bronchial wall on a target bronchus for BAL by using a SYNAPSE VINCENT volume analyser was the predicted factors for BAL recovery rate failure. For results in this study, it is considered that weakness of the bronchial wall might cause bronchial collapse during the BAL procedure. Therefore, we emphasized it in the manuscript, and we sincerely hope that the manuscript can provide new important information. 

Reviewer #2:

Dr Shimoda and al report the results of a retrospective study looking for predictors of poor bronchoalveolar lavage (BAL) recovery rate during bronchoscopy. A poor recovery rate was determined as below 30% of the instilled liquid.

Patients included were those, during the study period, who also had undergone thoracic CTscan allowing 3D image analysis of the bronchial tree.

Underlying conditions (COPD), tobacco use, age, lung lobe where the BAL was performed, cross-sectional area of the dependent bronchus and area of the bronchus wall were studied as potential predictors of BAL recovery rate.

The authors identified COPD, tobacco use history, BAL performed in another lobe than middle lobe or lingula and area of the bronchial wall as factors independently linked to the BAL recovery rate.

The authors used 3D image analysis to study the role of bronchial area and bronchial wall area; this makes the study very original.

Response: Thank you very much for your comment, and we are glad you reviewed our manuscript. 

I have few minor comments.

Though I am not a native English speaker, I wander if the manuscript doesn’t need English editing throughout.

Response: Thank you for your comment. The manuscript has been edited by a professional English-editing service, and the revised version has been also re-edited by a professional English-editing service. 

Introduction and methods: The authors should indicate clearly that an inclusion criterion was the availability of a thin-slice thoracic CT scan.

Response: Thank you for your comment. For a clear inclusion criterion, we added it in study design and setting on page 6, lines 100-101.

I am not sure that waiving the need for patient consent, even in a retrospective study, is really in accordance with the Declaration of Helsinki.

Response: Thank you for your comment. We approved the study by the Institutional Review Board of Fukujuji Hospital. This study was conducted retrospectively, and did not include any identifiable information for patients. Therefore, the Institutional Review Board decided waving the requirement for patients consent and we applied the opt-out method. We emphasized it on page 7, lines 111-113.

---

## [Decision Letter · Decision Letter 1]

2 Sep 2022

PONE-D-22-18216R1Analysis of Predicted Factors for Bronchoalveolar Lavage Recovery Failure: An Observational StudyPLOS ONE

Dear Dr. Shimoda,

Thank you for submitting your manuscript to PLOS ONE. After careful consideration, we feel that it has merit but does not fully meet PLOS ONE’s publication criteria as it currently stands. Therefore, we invite you to submit a revised version of the manuscript that addresses the points raised during the review process.

We look forward to receiving your revised manuscript.

Kind regards,

Taeyun Kim

Academic Editor

PLOS ONE

Journal Requirements:

Additional Editor Comments:

* My original comment was only deleting the "data.xlsx" which contains the individual information of all study participants, not all supplementary materials.

* I would like to suggest moving Table 2 to 1 as "Baseline characteristics between the failure group and success group", comparing all variables which was included in the Table 1 of your original submission. Then, Table 1 "Factors affecting the recovery rate of BAL" could be move to Table 2.

* In Table1, age is missing.

Reviewers' comments:

Reviewer's Responses to Questions

**Comments to the Author**

1. If the authors have adequately addressed your comments raised in a previous round of review and you feel that this manuscript is now acceptable for publication, you may indicate that here to bypass the “Comments to the Author” section, enter your conflict of interest statement in the “Confidential to Editor” section, and submit your "Accept" recommendation.

Reviewer #1: All comments have been addressed

Reviewer #2: (No Response)

2. Is the manuscript technically sound, and do the data support the conclusions?

Reviewer #1: Yes

Reviewer #2: Yes

3. Has the statistical analysis been performed appropriately and rigorously? 

Reviewer #1: Yes

Reviewer #2: Yes

4. Have the authors made all data underlying the findings in their manuscript fully available?

Reviewer #1: Yes

Reviewer #2: Yes

5. Is the manuscript presented in an intelligible fashion and written in standard English?

Reviewer #1: Yes

Reviewer #2: Yes

6. Review Comments to the Author

Reviewer #1: (No Response)

Reviewer #2: (No Response)

7. PLOS authors have the option to publish the peer review history of their article (what does this mean?). If published, this will include your full peer review and any attached files.

Reviewer #1: No

Reviewer #2: **Yes: **BOULAIN Thierry

---

## [Author Response · Author response to Decision Letter 1]

2 Sep 2022

Responses to the editor and the reviewers

Manuscript ID PONE-D-22-18216

“Analysis of Predicted factors for Bronchoalveolar Lavage Recovery Failure: A Case–Control Study”

All modifications have been highlighted in the revised manuscript.

To the editor and the reviewer:

Thank you very much for your constructive comments on our manuscript PONE-D-22-18216. We are pleased to hear that the reviewer found our study intriguing. In response to the journal requests and editer’s comments, we generated text and tables. We carefully studied all comments and made the necessary edits/modifications. The point-by-point responses are listed below. We hope that we have sufficiently addressed the issues raised by the reviewer.

Journal Requirements:

Response: Thank you for your comment. We ensured that all references are complete and correct. edited it on references. In addition, we modified them to “Vancouver” style.

Additional Editor Comments:

* My original comment was only deleting the "data.xlsx" which contains the individual information of all study participants, not all supplementary materials.

Response: Thank you for your comment. We re-uploaded two supplemental tables and one supplemental figure, and we described it in manuscript on page 12, line 201, page 15, lines 225-226, page 16, line 245, and on 26, lines 413-425. 

* I would like to suggest moving Table 2 to 1 as "Baseline characteristics between the failure group and success group", comparing all variables which was included in the Table 1 of your original submission. Then, Table 1 "Factors affecting the recovery rate of BAL" could be move to Table 2.

Response: Thank you for your comment. We changed table 1 to table 2, and table 2 to table 1. And the sentence that is related with tables changed the order.

* In Table1, age is missing.

Response: Thank you for your comment. Age and recovery rate are continuous variables; therefore, we evaluated the correlation between age and recovery rate by using Spearman’s correlation analysis. Please confirmed it on page 12, lines 200-203.

---

## [Decision Letter · Decision Letter 2]

13 Sep 2022

PONE-D-22-18216R2Analysis of Predicted Factors for Bronchoalveolar Lavage Recovery Failure: An Observational StudyPLOS ONE

Dear Dr. Shimoda,

Thank you for submitting your manuscript to PLOS ONE. After careful consideration, we feel that it has merit but does not fully meet PLOS ONE’s publication criteria as it currently stands. Therefore, we invite you to submit a revised version of the manuscript that addresses the points raised during the review process.

We look forward to receiving your revised manuscript.

Kind regards,

Taeyun Kim

Academic Editor

PLOS ONE

Journal Requirements:

Additional Editor Comments (if provided):

* I believe the authors have addressed almost everything the editor and reviewers raised. I have two minor comments regarding the construction of the Results section.

* Please correct "The comparisons between the failure group and 194 the success group are shown in Table 2". Comparisons were shown in Table 1. And this sentence would be better to be placed at the first sentence of the 2nd paragraph of Results section. In that paragraph 3rd sentence, please remove "In univariable analysis".

* Please separate the paragraph of Results section; one for supplementary Table 1 and another for main Table 2.

Reviewers' comments:

Reviewer's Responses to Questions

**Comments to the Author**

1. If the authors have adequately addressed your comments raised in a previous round of review and you feel that this manuscript is now acceptable for publication, you may indicate that here to bypass the “Comments to the Author” section, enter your conflict of interest statement in the “Confidential to Editor” section, and submit your "Accept" recommendation.

Reviewer #1: All comments have been addressed

2. Is the manuscript technically sound, and do the data support the conclusions?

Reviewer #1: (No Response)

3. Has the statistical analysis been performed appropriately and rigorously? 

Reviewer #1: (No Response)

4. Have the authors made all data underlying the findings in their manuscript fully available?

Reviewer #1: (No Response)

5. Is the manuscript presented in an intelligible fashion and written in standard English?

Reviewer #1: (No Response)

6. Review Comments to the Author

Reviewer #1: (No Response)

7. PLOS authors have the option to publish the peer review history of their article (what does this mean?). If published, this will include your full peer review and any attached files.

Reviewer #1: No

---

## [Author Response · Author response to Decision Letter 2]

14 Sep 2022

Responses to the editor and the reviewers

Manuscript ID PONE-D-22-18216

“Analysis of Predicted factors for Bronchoalveolar Lavage Recovery Failure: A Case–Control Study”

All modifications have been highlighted in the revised manuscript.

To the editor and the reviewer:

Thank you very much for your constructive comments on our manuscript PONE-D-22-18216. We are pleased to hear that the reviewer found our study intriguing. In response to the journal requests and editer’s comments, we generated text and tables. We carefully studied all comments and made the necessary edits/modifications. The point-by-point responses are listed below. We hope that we have sufficiently addressed the issues raised by the reviewer.

Journal Requirements:

Response: Thank you for your comment. We ensured that all references are complete and correct.

Additional Editor Comments:

* I believe the authors have addressed almost everything the editor and reviewers raised. I have two minor comments regarding the construction of the Results section.

* Please correct "The comparisons between the failure group and 194 the success group are shown in Table 2". Comparisons were shown in Table 1. And this sentence would be better to be placed at the first sentence of the 2nd paragraph of Results section. In that paragraph 3rd sentence, please remove "In univariable analysis".

Response: Thank you for your comment. We changed “The comparisons between the failure group and the success group are shown in Table 2.” to “in Table 1”. We moved “There were 36 patients (10.7%) in the failure group and 302 patients (89.3%) in the success group.” to the end of the 1st paragraph, and “The comparisons between the failure group and the success group are shown in Table 1” moved at the first sentence of the 2nd paragraph. Furthermore, "In univariable analysis" was removed.

* Please separate the paragraph of Results section; one for supplementary Table 1 and another for main Table 2.

Response: Thank you for your comment. We separated the paragraph related with Table 1 and Table 2.

---

## [Editor Report · Decision Letter 3]

15 Sep 2022

Analysis of Predicted Factors for Bronchoalveolar Lavage Recovery Failure: An Observational Study

PONE-D-22-18216R3

Dear Dr. Shimoda,

We’re pleased to inform you that your manuscript has been judged scientifically suitable for publication and will be formally accepted for publication once it meets all outstanding technical requirements.

Kind regards,

Taeyun Kim

Academic Editor

PLOS ONE
---

## [Editor Report · Acceptance letter]

21 Sep 2022

PONE-D-22-18216R3 

Analysis of Predicted Factors for Bronchoalveolar Lavage Recovery Failure: An Observational Study 

Dear Dr. Shimoda:

I'm pleased to inform you that your manuscript has been deemed suitable for publication in PLOS ONE. Congratulations! Your manuscript is now with our production department. 

Kind regards, 

on behalf of

Dr. Taeyun Kim 

Academic Editor

PLOS ONE